

# Landmark-based homologous multi-point warping approach to 3D facial recognition using multiple datasets

Olalekan Agbolade[1], Azree Nazri[1], Razali Yaakob[1], Abdul Azim Abd Ghani[2] and Yoke Kqueen Cheah[3]

[1] Department of Computer Science, Faculty of Computer Science & IT, Universiti Putra Malaysia, Serdang, Selangor, Malaysia
[2] Department of Software Engineering, Faculty of Computer Science & IT, Universiti Putra Malaysia, Serdang, Selangor, Malaysia
[3] Department of Biomedical Science, Faculty of Medicine and Health Sciences, Universiti Putra Malaysia, Serdang, Selangor, Malaysia

## ABSTRACT

Over the years, neuroscientists and psychophysicists have been asking whether data acquisition for facial analysis should be performed holistically or with local feature analysis. This has led to various advanced methods of face recognition being proposed, and especially techniques using facial landmarks. The current facial landmark methods in 3D involve a mathematically complex and time-consuming workflow involving semi-landmark sliding tasks. This paper proposes a homologous multi-point warping for 3D facial landmarking, which is verified experimentally on each of the target objects in a given dataset using 500 landmarks (16 anatomical fixed points and 484 sliding semi-landmarks). This is achieved by building a template mesh as a reference object and applying this template to each of the targets in three datasets using an artificial deformation approach. The semi-landmarks are subjected to sliding along tangents to the curves or surfaces until the bending energy between a template and a target form is minimal. The results indicate that our method can be used to investigate shape variation for multiple datasets when implemented on three databases (Stirling, FRGC and Bosphorus).

Corresponding author
Azree Nazri, azree@upm.edu.my

## INTRODUCTION

Human facial traits play an essential role in human identification. The face contains the most important sensory organs and acts as the central interface for appearance, communication, expression and mutual identification (*Peng et al., 2013*). Landmark-based geometric morphometric methods for face recognition provide new insights into patterns of biological shape variation that cannot be evaluated by traditional methods (*Anies et al., 2013*).

Recently, many two-dimensional face recognition systems have been developed, with good results for image acquisition under favorable conditions (*Zhao et al., 2003*). The

major constraints include illumination and changes in pose. In addition to variation due to pose and illumination, which affect 2D face data, 3D faces are more easily detected due to a higher intensity modality compared with 2D faces (*Savran, Sankur & Bilge, 2012*). Furthermore, when they are subjected to systematically increasing pitch and yaw rotation, as shown by *Wang et al. (2006)*, there is a drop in performance related to expression recognition in 2D, while that in 3D remains constant. This is a result of occlusion effects from substantial distortion in out-of-plane rotations. In addition, in regard to feature transformation and classification, the 3D modality shows some improvement over 2D with a high level of confidence. However, for depth features, both show the same performance. The processing cost of 3D models is higher than that of 2D models (*Savran, Sankur & Bilge, 2012*).

The term 'morphometrics' was coined more than 50 years ago by Robert E. Blackith, who applied multivariate statistical methods to the basic carapace morphology of grasshoppers (*Elewa & Elewa, 2010*). Morphometrics is the study of shape variation and its covariation with other variables (*Bookstein, 1997a*; *Dryden & Mardia, 1998*). According to *Adams, Rohlf & Slice (2004)*, the term traditionally referred to the application of multivariate statistical analyses to sets of quantitative variables such as length, width, height and angle. Advances in morphometrics have shifted the focus to the Cartesian coordinates of anatomical points that can be used to define more traditional measurements. Morphometrics considers variation and group differences in shape, central tendency, and the association of shape with extrinsic factors. This is directly based on the digitized $(x, y, z)$-coordinate positions of landmarks, or points representing the spatial positions of putatively homologous structures in 2D or 3D. In contrast, conventional morphometric studies utilize distances as variables (*Bookstein, 1997a*; *Dryden & Mardia, 1998*; *Rohlf, 1993*).

The thin-plate spline (TPS) is simply a convenient function for capturing changes in landmark configurations and displaying the differences on the smoothest possible transformation grid (*Rohlf, Loy & Corti, 1996*). This ensures that the points of the starting and target form appear precisely in their corresponding positions in relation to the transformed and untransformed grids (*Bookstein, 1989*). With the application of the iterative closest point (ICP), landmark correspondence can be iteratively registered in the vicinity of a landmark with a re-weighted error function. In work by *Wan et al. (2010)*, a thin-plate spline (TPS) was used to align the points in each facial image. These authors employed ICP to build a correspondence by taking the closest point on each surface mesh, while the inverse of the TPS warp was used to map each surface back to its reference location. The smoothness or point relaxed in TPS was approached based on the minimization of bending energy. This approach computes the amount of deformation between two shape configurations, as quantified by the TPS function through the integral of the squared second derivatives of that deformation (*Mitteroecker & Gunz, 2009*). The geometric morphometrics (GM) of the Procrustes superimposition method is a least-squares oriented method involving translation, scaling and rotation (*Mitteroecker & Gunz, 2009*). Shape is the geometric information of an object after the removal of location, orientation and scale (*Kendall, 1977*).

The use of GM has revolutionized the sophistication of the collection and quantitative analysis of biological shapes. It has been applied to solve various research questions relating to plants, animals and humans. Examples include Neanderthal fossils (*Rosas et al., 2015*), flower shapes (*Van der Niet et al., 2010*), dinosaurs (*Fearon & Varricchio, 2015*), butterfly wings (*Chazot et al., 2016*), zebrafish skeletogenesis (*Aceto et al., 2015*) and humans (*Solon, Torres & Demayo, 2012*; *Lindner et al., 2016*).

A landmark was defined by *Marcus et al. (1993)* as a point in a 2D or 3D space that corresponds to the position of a particular trait in an object. *Dryden & Mardia (1998)* also described landmarks as points of correspondence on each object that match within and between populations. This set of points, one on each form, are operationally defined for an individual based on local anatomical features, and must be consistent with some hypothesis of biological homology. *Bookstein (1997a)* and *Dryden & Mardia (1998)* categorized landmarks into three types: Types I, II and III. Type I landmarks are defined as discrete juxtapositions of tissues such as at the intersection of three sutures, for example the dacryon and asterion (*Lynch, Wood & Luboga, 1996*; *Slice, 2006*) or the bregma and lambda (*Bookstein, 1997b*). Type II landmarks are curvature maxima or other local morphogenetic processes, usually with a biomechanical implication such as a muscle attachment site (*Ross & Williams, 2008*), for example the prosthion, ectoconchion (*Lynch, Wood & Luboga, 1996*; *Slice, 2006*), subnasale, meatus or nasion (*Bookstein, 1997b*). Type III landmarks are extremal points such as the endpoints of maximum cranial length and breadth (*Lynch, Wood & Luboga, 1996*; *Slice, 2006*) and the orbitale, gonion, glabella and gnathion (*Bookstein, 1997b*).

Facial landmarking is a crucial step in facial analysis for biometrics and numerous other applications. Since 3D data contain more information and are less sensitive to illumination and occlusion than 2D data, the use of 3D data to improve facial analysis is increasing in computer vision (*Chen et al., 2015*). Many studies of population variation have been performed in morphometric research using facial landmarks. Some investigations have covered certain regions of the face, while some have examined the entire facial region. For instance, significant differences in the symmetric shape component in the nasal region between German and Chinese populations were identified by *Schlager & Rüdell (2015)*, using a dense set of semi-landmarks. An investigation of nose profile morphology in Scottish and Indonesian populations was also carried out by *Sarilita et al. (2018)*, with the aim of improving the accuracy of forensic craniofacial reconstruction. The soft-tissue facial form of high-resolution 3D images of Han Chinese, Tibetan, Uyghur and European people was analyzed by *Guo et al. (2014)*. Facial diversity was examined by establishing a high-density alignment across all faces, and the analyses revealed that the brow area, cheekbones, and nose exhibited strong signals of differentiation between populations. Statistically significant face shape differences between the Dutch and UK population were investigated by *Hopman et al. (2014)*. Mean face shape was visualized using signature heatmap and dynamic morphs, showing that genetic variants influence normal facial variation. Although these studies examined variation in diverse populations, these populations were drawn from the same datasets. In contrast, the current study analyzes variation using different datasets.

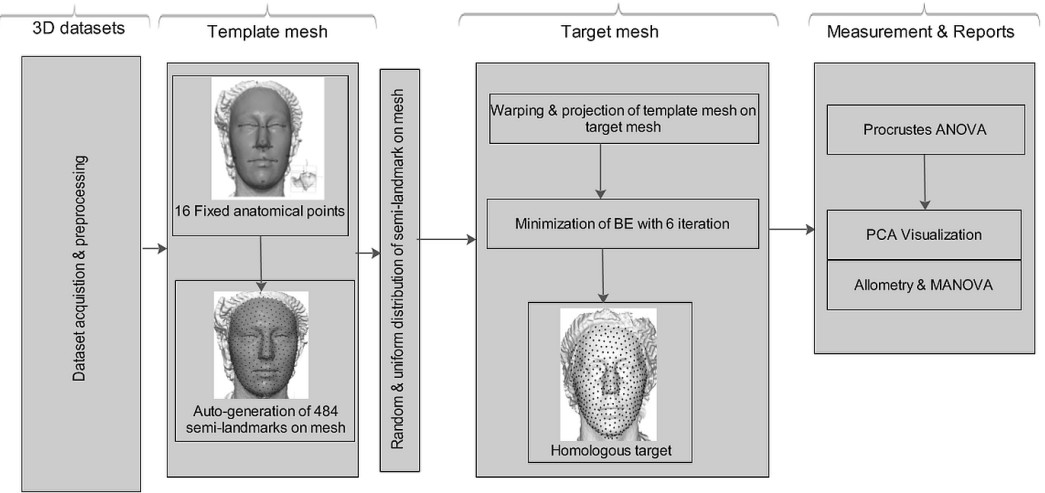

**Figure 1** Schematic conceptual diagram of the proposed homologous multi-point warping algorithm.

Since wide biological variability cannot be assessed using only anatomical landmarks (*Botton-Divet et al., 2015*), in order to quantify complex shapes, sliding semi-landmarks have been developed which can be placed on surfaces (*Gunz, Mitteroecker & Bookstein, 2005*) or curves (*Bookstein, 1997a*; *Gunz, Mitteroecker & Bookstein, 2005*). This approach generates landmarks that are spatially homologous after sliding (*Parr et al., 2012*) and can be optimized by minimizing the bending energy (*Cornette et al., 2013*; *Fabre et al., 2014*) or Procrustes distance (*Perez, Bernal & Gonzalez, 2006*; *Mitteroecker et al., 2013*). Several software packages are currently available to perform the sliding of landmarks in 3D, for example Edgewarp (*Bookstein & Green, 1994*), the EVAN toolbox ( http://evan-society.org), Viewbox (*Halazonetis, 2014*), Mathematica (*Mitteroecker et al., 2013*), and the geomorph (*Adams & Otárola-Castillo, 2013*) and Morpho (*Schlager, 2013*) R packages.

The aim of this study is to apply the computational deformation process reported by *Bookstein (1989)*. When projecting a surface semi-landmark from the template object to the target object and iteratively sliding the semi-landmark to a point relaxed, the simpler workflow allows us to perform this task in Viewbox 4.0, unlike the complex workflow presented by *Botton-Divet et al. (2015)*. Secondly, this method is not new in terms of analyzing shape variation in morphometric geometry, but its application to the analysis of shape variation for soft-tissue faces in 3D for multiple human datasets is novel. Figure 1 shows a schematic conceptual diagram of the homologous multi-point warping algorithm.

## MATERIALS & METHODS

The use of 3D face images in morphometrics not only gives us scope to cover a wider area of the human facial region, but also retains all the geometric information of the object descriptors (*Bookstein, 1997b*; *Dean, 1996*). Here, our method uses 3D facial images from

three different datasets, and uses a morphometric approach to propose a less mathematically complex yet robust algorithm for facial landmarks in 3D.

## Dataset & description

We used three datasets to validate the robustness of our method. The first was acquired from the Stirling/ESRC 3D face database, which was captured by a Di3D camera system (*Stirling-ESRC, 2018*). These images are in the format of wavefront obj files containing 101 subjects with 3D facial scans in a neutral position. The database was intended to facilitate research into face recognition, expression and perception, and we randomly selected 58 subjects for this study. The dataset was used as a test set for a competition involving 3D face reconstruction from 2D images, with the 3D scans acting as the 'ground truth', at an IEEE conference. The second dataset was the Bosphorus database, which was intended for research on 3D and 2D human face processing tasks and contains 105 subjects. We randomly selected 57 subjects for this study. The dataset was acquired using structured-light-based 3D system, with the subjects being instructed to sit at a 1.5 m distance from the camera; the sensor resolution in the x, y and z directions was 0.3, 0.3, and 0.4 mm, respectively, and high-resolution color texture was used (*Savran et al., 2008*). The third dataset was the Face Recognition Grand Challenge Version 2.0 (FRGC v2) database, consisting of 466 facial images, of which we randomly selected 120 subjects for this study. Each subject image was captured under uniform illumination, with high resolution and fairly uncontrolled conditions (*Phillips et al., 2005*).

## Template mesh

A template mesh of 92,995 vertices and 183,996 triangles was created by manually locating 16 anatomical points on the 3D face (Fig. 2) called fixed points, according to facial landmark standards (*Caple & Stephan, 2016*) (for more detail, see Table 1). The fixed landmarks were not subjected to sliding, but were used to establish the warping fields for minimizing the bending energy. Due to its ease of detection and pose correction (*Creusot, Pears & Austin, 2010*) and its invariance to facial expression (*Colombo, Cusano & Schettini, 2006*), the nose tip (pronasale) was selected as the most robust and prominent landmark point. The nose tip area can be approximated as a hemisphere on the human face, although any other facial anatomical point could be used. This is where the sliding points begin to spread across the facial surface. Using this fixed point (the pronasale), 484 semi-landmarks were automatically generated, with the overlapping on the pronasale shown in blue. These were first randomly placed on the facial mesh before being uniformly distributed on the selected facial surface (Fig. 3). This was done using the locational positions of the fixed anatomical points with 1.5 mm radius to accommodate all 500 points. To quantify the morphological data for the complex 3D traits of both reference and target shapes, we used geometric morphometric tools based on previously reported landmark-based methodologies (*Halazonetis, 2014*; *Klingenberg & Zaklan, 2000*; *Kouli et al., 2018*; *Yong et al., 2018*; *Zelditch, Swiderski & Sheets, 2012*), and the automatic point placement, semi-landmark sliding, and landmark acquisition were implemented in ViewBox 4.0 (*Halazonetis, 2014*).

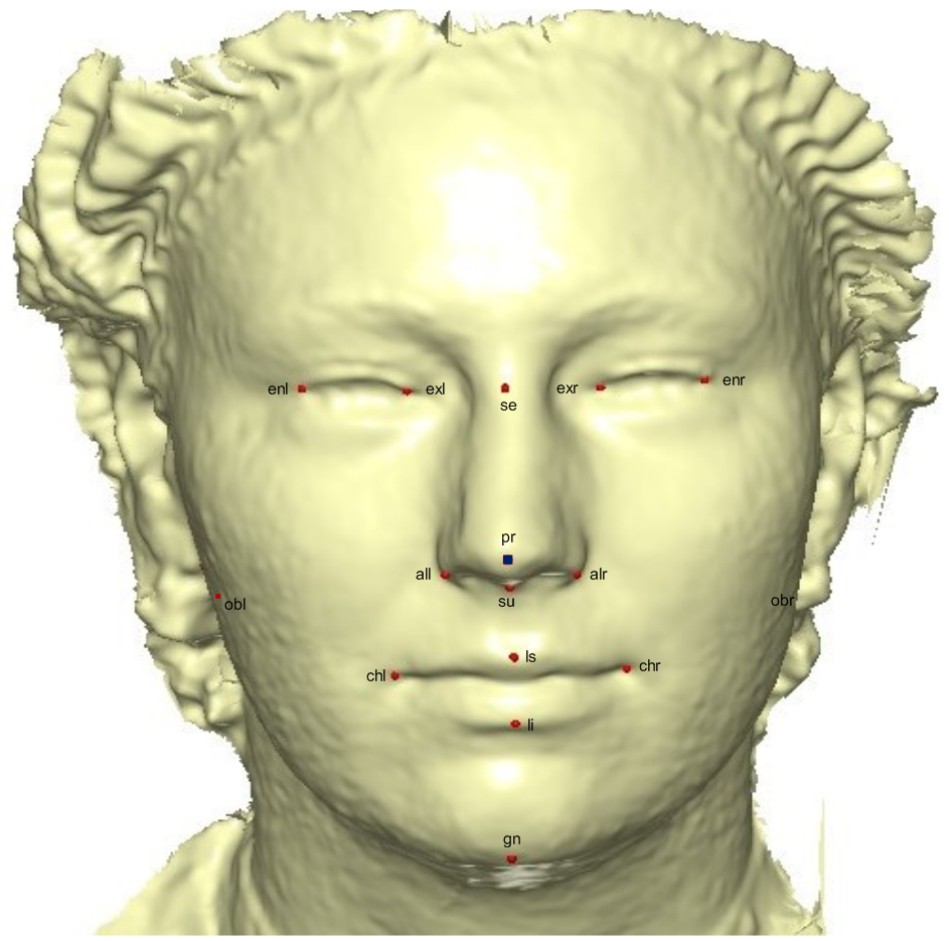

**Figure 2** **A 3D mesh template with the location of the prominent point at the center of the face for pose-invariant correction.** The 16 fixed anatomical landmarks are shown in red, while the blue area on the pronasale indicates the point at which the semi-landmarks begin the sliding process.

## Homologous multi-point warping

The geometry of curves and surfaces is easy in 2D or 3D, but it is less easy to define semi-landmarks for non-planar surfaces in 3D (*Huanca Ghislanzoni et al., 2017*) as they are not guaranteed to be homologous after the first placement. However, this could be achieved by subjecting the semi-landmarks to sliding in the direction that reduces shape variance, thus closely positioning the points at the same locations in the 3D space. The sliding step is important, as it places the landmarks in positions where they correspond better to each other between individuals (*Mitteroecker et al., 2013*). These semi-landmarks were allowed to slide on the curves and the surface mesh of each target using TPS warping of the template, which positioned the reference points on the target facial mesh by minimizing the bending energy.

According to *Bookstein (1989)*, physical steel takes a bending form with a small displacement. This is because the function $(x, y, z)$ is the configuration of lowest physical bending energy, which is consistent with the given constraints. In this 3D face deformation,

**Table 1  Fixed anatomical landmarks and descriptions.**

| No | Fixed Landmarks | 3D Notation | Description |
|---|---|---|---|
| 1 | Endocanthion left | enl | Left most medial point of the palpebral fissure, at the inner commissure of the eye |
| 2 | Exocanthion left | exl | Left most lateral point of the palpebral fissure, at the outer commissure of the eye |
| 3 | Exocanthion right | exr | Right most lateral point of the palpebral fissure, at the outer commissure of the eye |
| 4 | Endocanthion right | enr | Right most medial point of the palpebral fissure, at the inner commissure of the eye |
| 5 | Sellion | se | Deepest midline point of the nasofronal angle |
| 6 | Pronasale | pr | The most anteriorly protruded point of the apex nasi |
| 7 | subnasale | su | Median point at the junction between the lower border of the nasal septum and the philtrum area |
| 8 | Alare left | all | Left most lateral point on the nasal ala |
| 9 | Alare right | alr | Right most lateral point on the nasal ala |
| 10 | Cheilion left | chl | Left outer corners of the mouth where the outer edges of the upper and lower vermilions meet |
| 11 | Cheilion right | chr | Right outer corners of the mouth where the outer edges of the upper and lower vermilions meet |
| 12 | Labiale superius | ls | Midpoint of the vermilion border of the upper lip |
| 13 | Labiale inferius | li | Midpoint of the vermilion border of the lower lip |
| 14 | Gnathion | gn | Median point halfway between pogonion and menton |
| 15 | Obelion left | obl | Left median point where the sagittal suture intersects with a transverse line connecting parietal foramina |
| 16 | Obelion right | obr | Right median point where the sagittal suture intersects with a transverse line connecting parietal foramina |

the transformation of TPS is done mathematically via the interpolation of a smooth mapping of $h$ from $\mathbb{R}^3 \rightarrow \mathbb{R}^3$. This is a selected set of corresponding points $P_{Ri}, P_{Ti}$, $i = 1, \ldots, N$ on the faces of the reference object (template) and target (subject) that minimizes the bending energy function $E(h)$ using the following interpolation conditions (*Bookstein, 1989*; *Bookstein, 1997a*; *Corner, Lele & Richtsmeier, 1992*):

$$E(h) = \iiint_{\mathbb{R}^3} \left( \left(\frac{\partial^2 h}{\partial x^2}\right)^2 + \left(\frac{\partial^2 h}{\partial y^2}\right)^2 + \left(\frac{\partial^2 h}{\partial z^2}\right)^2 + 2\left(\frac{\partial^2 h}{\partial xy}\right)^2 + 2\left(\frac{\partial^2 h}{\partial xz}\right)^2 + 2\left(\frac{\partial^2 h}{\partial yz}\right)^2 \right)$$
$$dxdydz, s.t. \ h(P_{Ti}) = P_{Ri}, i = 1, \ldots, M \tag{1}$$

where $P_{Ti}$ is the target object, $P_{Ri}$ is the reference object for the sets of corresponding points, and $h$ is the bending energy function that minimizes the non-negative quantity of the interpolation of the integral bending norm or the integral quadratic variation $E(h)$. The TPS method now decomposes each component into affine and non-affine components, such that

$$h(P_h) = \Psi(P_h)K + P_h\Gamma \tag{2}$$

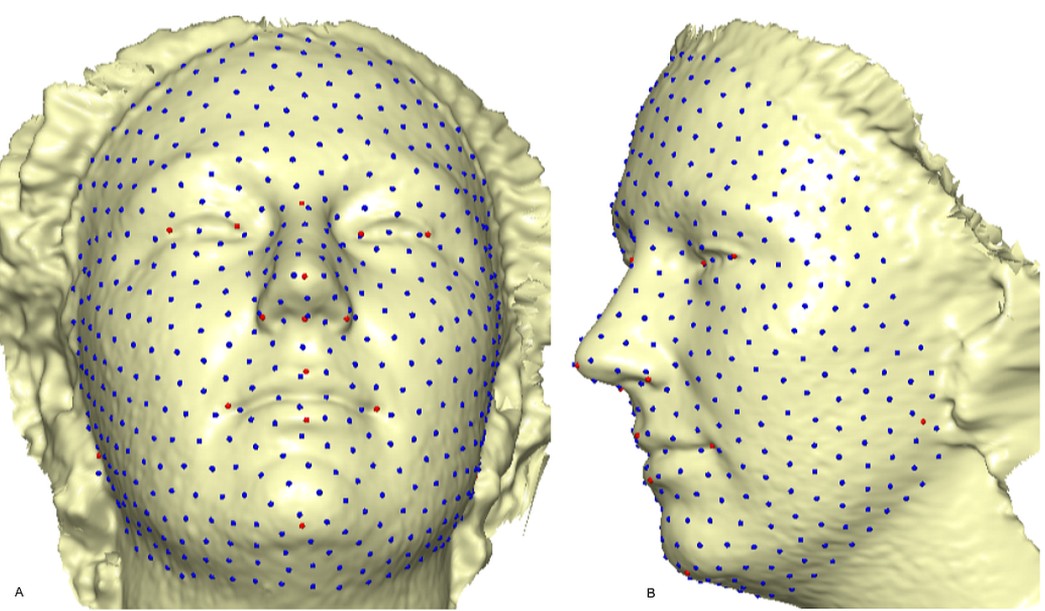

**Figure 3** **A 3D mesh template of the reference model with 500 landmarks.** Showing 16 fixed anatomical points and 484 semi-landmarks with 1.5 mm radius: (A) frontal skewed view; (B) profile view.

where $P_h$ are the homogeneous coordinate points on the target 3D face, and $\Psi(P_h) = \Psi_1(P_h), \Psi_2(P_h), \dots, \Psi_M(P_h)$ is a $1 \times M$ kernel vector of TPS of the form:

$$\Psi_w(P_h) = \parallel P_h - P_{Tw} \parallel \tag{3}$$

K is a $M \times 4$ non-affine warping coefficient matrix, and $\Gamma$ is a homogeneous affine transformation of a $4 \times 4$ matrix. The energy function is minimized to find the optimum solution to Eq. (4) if the interpolation condition in Eq. (1) is not met.

$$\mathbb{E}(\beta, K, \Psi) = \frac{1}{M} \sum_{J=1}^{M} \parallel h(P_{Tj}) - P_{Rj} \parallel + \beta E(h). \tag{4}$$

The interpolation conditions in Eq. (1) are satisfied if the smoothing regularization term $\beta$ is zero, where $\Gamma$ and K are TPS parameters obtained by solving the linear equation:

$$\begin{pmatrix} \Psi & P_R \\ P_R^T & 0 \end{pmatrix} \begin{pmatrix} K \\ \Gamma \end{pmatrix} = \begin{pmatrix} P_T \\ 0 \end{pmatrix} \tag{5}$$

$\Psi$ is a $M \times M$ matrix with components $\Psi_{wl} = \parallel P_{Tw} - P_{Tl} \parallel$ and $P_R$ is a $M \times 4$ matrix in which each row is the homogeneous coordinate of the point $P_{Ri}, i = 1, \dots, M$. Using Eq. (2), the target facial mesh $P_{Ti}$ was deformed to the reference mesh $P_{Ri}$. The bending energy was applied, and the process was iterated for six cycles to achieve optimum sliding of the points on the facial surface which gives points relaxed. This changed the bending energy from the initial value $E_i$ to the final value $E_f$ after six complete iterations. This means that the semi-landmarks can be treated in the same way as homologous landmarks in downstream analyses. Since warping may result in points that do not lie directly on the

facial surface on the target mesh, the transferred points were projected onto the closest point on the mesh surface using the ICP method (*Creusot, Pears & Austin, 2010*). The aim of using ICP is to iteratively minimize the mean square error between two point sets. If the distance between the two points is within an acceptable threshold, then the closest point is determined as the corresponding point (*Mian, Bennamoun & Owens, 2008*). The homologous landmark warping $H_{K\Gamma}$ after six complete iterations is therefore:

$$H_{K\Gamma} = E_{f-i} \begin{pmatrix} K \\ \Gamma \end{pmatrix} \tag{6}$$

where

$$\begin{pmatrix} K \\ \Gamma \end{pmatrix} = \begin{pmatrix} \Psi & P_R \\ P_R^T & 0 \end{pmatrix}^{-1} \begin{pmatrix} P_T \\ 0 \end{pmatrix} \tag{7}$$

is the linear TPS equation obtained during deformation of the surface of the target mesh to the reference mesh, before convergence was finally reached, and $E_{f-i} = E_f - E_i$ after six complete iterations. The first iteration showed a partial distribution of sliding points on the target surface mesh (Fig. 4A). This was automatically repeated until the optimum homologous result was achieved, using an exponential decay sliding step of hundred to five percent. During relaxation of the spline, the semi-landmarks were slid along the surface and the curve tangent structures, rather than on the surfaces or the curves. This reduced the computational effort, as the minimization problem became linear since sliding along the tangents lets the semi-landmarks slip off the data (*Gunz, Mitteroecker & Bookstein, 2005*). The target surface mesh was then treated as a set of homologous points (Fig. 4B). Note that we did not construct a new deformable mathematical equation from scratch, but simply extended the standard deformable method established by *Bookstein (1989)*. We added a minor extension to the computational process of projecting the surface semi-landmark from the template object to the target object and iteratively sliding the semi-landmark to a point relaxed.

After applying the step-by-step methods of digitization of the facial points and sliding of the semi-landmarks using ViewBox 4.0, we then applied Procrustes superimposition, error assessment with Procrustes ANOVA, canonical variate analysis, regression analysis, and shape visualization and variation with PCA, using MorphoJ 1.06d (*Klingenberg, 2011*). Boxplots for the size distribution and a MANOVA were performed in PAST 3.0 (*Hammer, Harper & Ryan, 2001*).

## Sliding task comparison

In many studies dealing with sliding semi-landmarks, many iterations and tasks are required before optimum smoothness is reached, even with a small set of landmarks. *Botton-Divet et al. (2015)* compared Edgewarp (*Bookstein & Green, 1994*) and Morpho (*Schlager, 2013*) in terms of the time, workflow complexity, and computational efficiency required to slide semi-landmarks on long bones in different mustelids. The study used 27 manually placed 3D anatomical landmarks and 790 sliding semi-landmarks on curves and surfaces. Our study does not consider the time factor, due to the challenges arising from the different

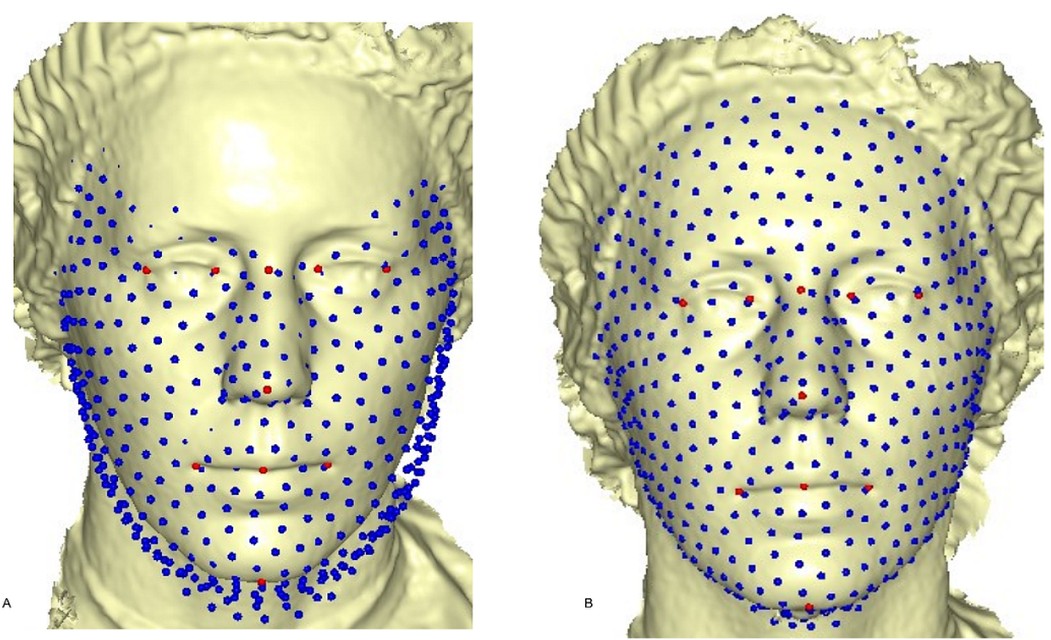

**Figure 4  Sliding point warped on the surface of the target face.** (A) partial sliding on target mesh; (B) complete and homologous warping on target mesh.

datasets and samples used. Furthermore, many studies that have implemented sliding semi-landmarks have not reported information on the sliding times, giving no scope for comparison. In our study, both the digitization of fixed landmarks and the sliding of semi-landmarks were performed in Viewbox. Unlike in Edgewarp, where digitization was first performed using a Breuckmann 3D white light fringe surface scanner (*Botton-Divet et al., 2015*), the remaining holes were filled and edges and spikes were removed using Geomagic. It should be noted that the task of surface pre-processing is not required in Viewbox, unlike in Edgewarp, where surface pre-processing, initial projection, and iterations against the Procrustes consensus must be accounted for (*Botton-Divet et al., 2015*).

## Measurement error

The process of landmark coordinate extraction is always associated with some degree of measurement error, and this may be as a result of the non-coplanarity of landmarks, inconsistency of specimens relative to the plane of digitization, or difficulties in pinpointing the landmark locus (*Webster & Sheets, 2010*). Landmark digitization error can be minimized by careful landmark selection, but can never be totally eliminated. In assessing measurement error, three templates were designed for each population. The same individuals (five per dataset) were acquired three times each, using the three templates (*Klingenberg, Barluenga & Meyer, 2002*; *O'Higgins & Jones, 1998*; *Robinson & Terhune, 2017*). These were digitized for both manual and sliding semi-landmarks on the different reference objects, and this was followed by Procrustes superimposition on the landmark data using three partitions: fixed anatomical landmarks (FAL), sliding semi-landmarks (SSL), and combined landmarks

(CL). Although several other error measurement methods were suggested by *Fruciano (2016)*, the measurement error for this study was assessed using a Procrustes ANOVA. This technique (*Klingenberg, Barluenga & Meyer, 2002*; *Klingenberg & McIntyre, 1998*) was implemented in morphometrics to analyze measurement errors (*Klingenberg et al., 2010*; *Leamy et al., 2015*; *Singh et al., 2012*) using MorphoJ, which was achieved through the minimization of the squared sum of the distances of all objects and the consensus configuration (*Fruciano, 2016*). This approach uses a partition based on the sum of squares of the deviations from the average configuration of each coordinate in a two-factor ANOVA, which can then be summed across all the coordinates. Using a relevant number of degrees of freedom, computation of the mean squares is done by dividing the total sum of squares for an effect (*Klingenberg & McIntyre, 1998*). The Procrustes approach is most useful in computing mean shapes and in deciding whether two shapes $[R_i]$ and $[T_i]$ are random realizations of the same shape (*Goodall, 1991*).

The steps in this algorithm can be summarized as follows:

1. Anatomical fixed points (16) were digitized on the template facial mesh and a prominent point (the pronasale) was identified.
2. Semi-landmarks (484) were automatically generated and placed along the curves, located at a uniform distance along each curve for sliding in Step 5.
3. These semi-landmarks were first randomly placed and then uniformly distributed on the selected reference surface mesh, starting from the selected prominent point.
4. The reference facial model was warped to each target mesh configuration using a TPS transformation, and the surface semi-landmark was projected from the reference facial mesh to the target facial mesh.
5. The surface and curve semi-landmarks were then slid together in the direction that minimized the bending energy between each target configuration and the reference object. This was done iteratively in six complete cycles, in order to ensure convergence and optimum smoothness. This gave a homologous representation of the reference mesh.
6. A Procrustes superimposition of the landmark data was performed, and an error assessment was computed using a Procrustes ANOVA.

## Shape and size variation

Since there may be an interaction between the size and shape in facial morphology due to changes in the shape associated with size differences (*Klingenberg & McIntyre, 1998*), we assessed the allometry by testing the statistical significant proportion of morphological variation in the shape components, using a multivariate regression of shape onto size. Canonical variate analysis (CVA) was also performed to test the group differences and an ordination plot was produced (plot not shown).

Differences in the effects and size were then examined by computing a non-parametric analysis of variance (MANOVA) in terms of Wilks' lambda. Using the population as the group and the size as the covariate, the population by size interaction term was calculated. The MANOVA was recomputed after removing the interaction term (population by size) and the population effect tested for the difference in the regression intercept.

# RESULTS

## Sliding task

The tasks required for the initial projection and the relaxations against the Procrustes mean shape for the iterations are minimal in Viewbox, which requires a much simpler workflow compared to Edgewarp. Moreover, since a general Procrustes analysis was required in order to perform the sliding task, which is not implemented internally in Edgewarp, new input files were generated prior to performing the next iteration (*Botton-Divet et al., 2015*), creating greater complexity in the workflow.

## PCA

After calculating the mean shape using Procrustes superimposition, we studied the variability in the shape using PCA. This was not only to reduce the number of dimensions but also to balance the fit of the model with the ease of analysis and potential loss of information (*Cangelosi & Goriely, 2007*). The PCA of the total sample yielded 168 principal components. When they were separately computed, the Stirling population yielded only 57 PCs, FRGC yielded 59 PCs, and Bosphorus 51 PCs, all with non-zero variability. In order to retain any component that accounted for a specific proportion or percentage, we used a broken stick approach to PCA selection (*Cangelosi & Goriely, 2007*; *Klingenberg, 2013*; *Peres-Neto, Jackson & Somers, 2005*). The first five PCs accounted for 98.05% of the total variation in the Stirling population, while the first four PCS accounted for 93.44% of the total variation in the FRGC population, the first three PCs accounted for 85.65% of the total variation in the Bosphorus population, and the first two PCs accounted for 97.17% of the total variation in the combined population (*Cangelosi & Goriely, 2007*). The first and second PCs (PC1 and PC2) of the Stirling population accounted for 65.40% and 20.03% of the variation, respectively; in the FRGC population, these accounted for 76.88% and 7.97%, respectively; for the Bosphorus population, these figures were 68.29% and 10.84%, respectively; and for the total population, they were 93.96% and 3.21%, respectively.

Shape variations for each population and the combined population are shown in Fig. 5. In the visualization of 3D dataset object, the number of landmarks is shown in red and the mean symmetry configuration is shown in light blue. A lollipop graph is shown of the first principal component of each population and the combined population, which indicates the difference in the face shape. Note that we only visualized the first PC of each population, since this accounted for the highest variation in the total shape.

## Procrustes ANOVA and shape variation

A Procrustes superimposition of each set configuration produced a symmetric consensus configuration. When the variation was partitioned around this consensus using a Procrustes ANOVA (Table 2), the results showed that the variation in symmetric shape among individuals in each set accounted for the largest portion of the total variation, although this was not statistically significant for the manually placed landmarks. To assess the digitization errors of the manually placed landmarks, the sliding semi-landmarks and overall landmarks, the deviations for each were obtained by simply calculating the amount of displacement from the average position calculated from all digitization and the variation accounts for the

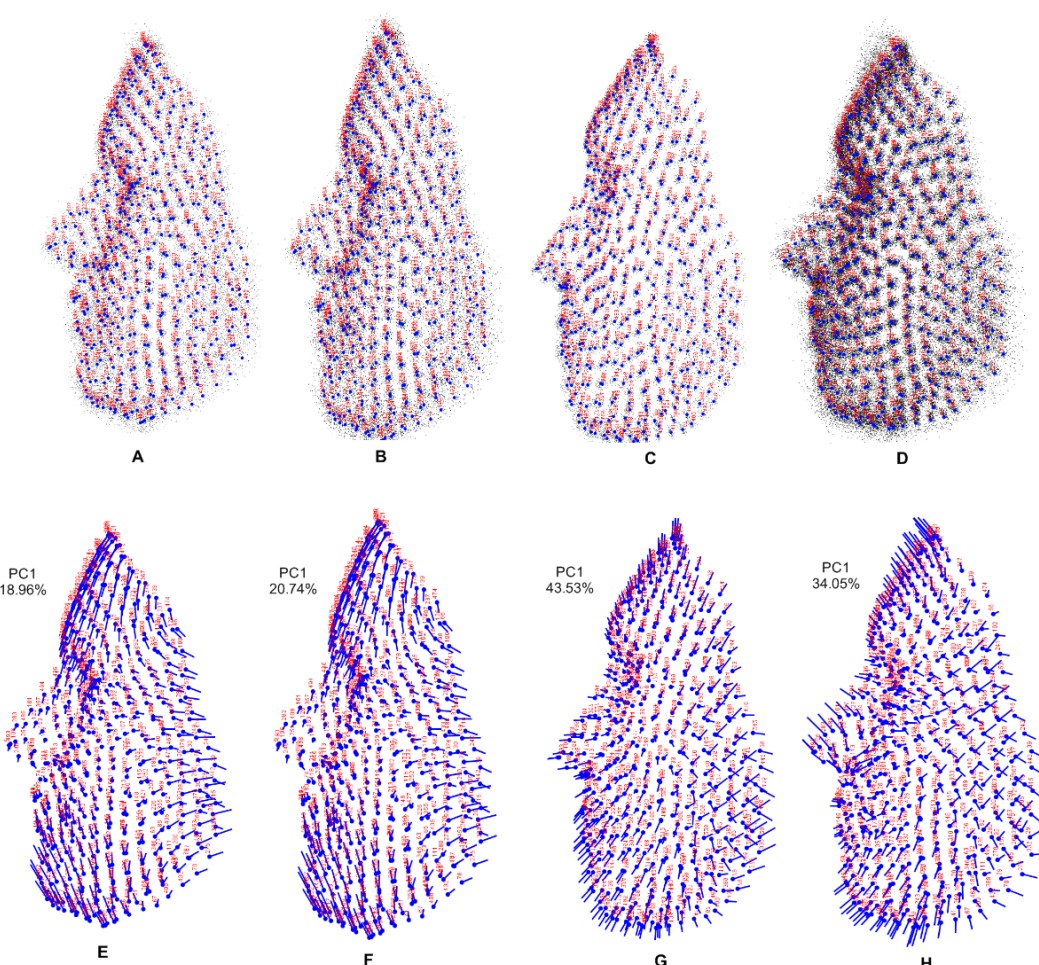

**Figure 5** **Visualization using 3D dataset objects and lollipop graphs.** (A–D) represent a 3D dataset object on three axes, showing only axis 1 vs axis 3: (A) Stirling; (B) FRGC; (C) Bosphorus; (D) combined population, with variance contributed by each PC. (E–H) show lollipop visualizations of the first principal component of each population and combined population: (E) Stirling; (F) FRGC; (G) Bosphorus; (H) combined population.

smallest portion of the total variation. For manually placed (fixed anatomical) landmarks, the variation accounted for 4.31%, while for sliding semi-landmarks, this accounted for 3.71%, and for the overall landmarks, this was 3.67%.

The CVA indicated that each population studied was clearly distinct from the others. The Procrustes distances between populations (FRGC vs. Bosphorus $= 0.040$; Stirling vs. Bosphorus $= 0.580$; Stirling vs. FRGC $= 0.061$) were all statistically significant ($p < 0.0001$). All 10,000 pairwise permutation tests indicated that the mean shapes differed significantly in the population.

Figure 6A shows the first two principal components of the total shape in the three populations, accounting for the total variance. The scatterplots of the scores along the first two principal components for all the datasets (Fig. 6A) showed an apparent pattern

**Table 2** **Procrustes ANOVA for facial shape with digitization errors.** Top: fixed anatomical landmarks (FAL) only; centre: sliding semi-landmarks (SSL); bottom: combined landmarks (CL).

| Effect | Var explained (%) | SS | MS | DF | F | P |
|---|---|---|---|---|---|---|
| Population | 25.82 | 0.376064 | 0.004586 | 82 | 42.86 | <.0001 |
| Individuals | 69.87 | 1.017737 | 0.000107 | 9512 | 1.05 | 0.2239 |
| Error (FAL) | 4.31 | 0.062844 | 0.000102 | 615 | | |
| Total | 100 | 1.456644 | 0.004795 | 10209 | | |
| | | | | | | |
| Population | 30.41 | 0.215282 | 0.0000744922 | 2890 | 53.56 | <.0001 |
| Individuals | 65.88 | 0.466278 | 0.0000013909 | 335240 | 1.15 | <.0001 |
| Error (SSL) | 3.71 | 0.026243 | 0.0000012108 | 21675 | | |
| Total | 100 | 0.707804 | 0.0000770939 | 359805 | | |
| | | | | | | |
| Population | 29.48 | 0.218988 | 0.0000733384 | 2986 | 51.15 | <.0001 |
| Individuals | 66.85 | 0.496599 | 0.0000014337 | 346376 | 1.18 | <.0001 |
| Error (CL) | 3.67 | 0.027269 | 0.0000012176 | 22395 | | |
| Total | 100 | 0.742857 | 0.0000759897 | 371757 | | |

**Notes.**

SS, sum of squares; %Var, percentage of variance; MS, mean square; DF, degrees of freedom; F, Fstatistic; P, $P$-value (parametric).

of association between the manual anatomical landmarks and the sliding semi-landmarks for the same specimen. This pattern disappeared when the sliding semi-landmarks were removed, as the data for Bosphorus and FRGC were clustered more tightly.

Changes in the shape of the face as a function of size (allometry) were evaluated using multivariate regression of the effect of shape variables on the centroid size for all populations (Fig. 6B). This is usually of primary interest within a homogeneous population. Inter-population allometry explained only 13.28% of the shape differences according to size, and was significant ($p < 0.0001$). The box-plots in Fig. 6C showed that the Bosphorus dataset contained the largest sizes, followed by FRGC, while the Stirling dataset contained the smallest sizes.

The characteristics of the allometric trajectories of the population were then tested using a MANOVA (Table 3), which explained a significant proportion of the overall variation. The interaction term (test for slopes) was statistically significant. When the size effect was removed and the MANOVA was repeated, the result was still statistically significant, suggesting that the effect of size on shape for both the slope and intercept was strong, and that this was not the case in the population group.

## DISCUSSION

The use of landmarks has evolved in terms of locating biological or anatomical features on human faces. However, its validity is based on a morphometric analysis, which depends upon the biological justification for the designation of the landmarks, as stated by *Bookstein (1997b)*. In performing the sliding task, Edgewarp appears to be more task-complex than Viewbox. The iterative relaxation in Edgewarp requires several manual operations per

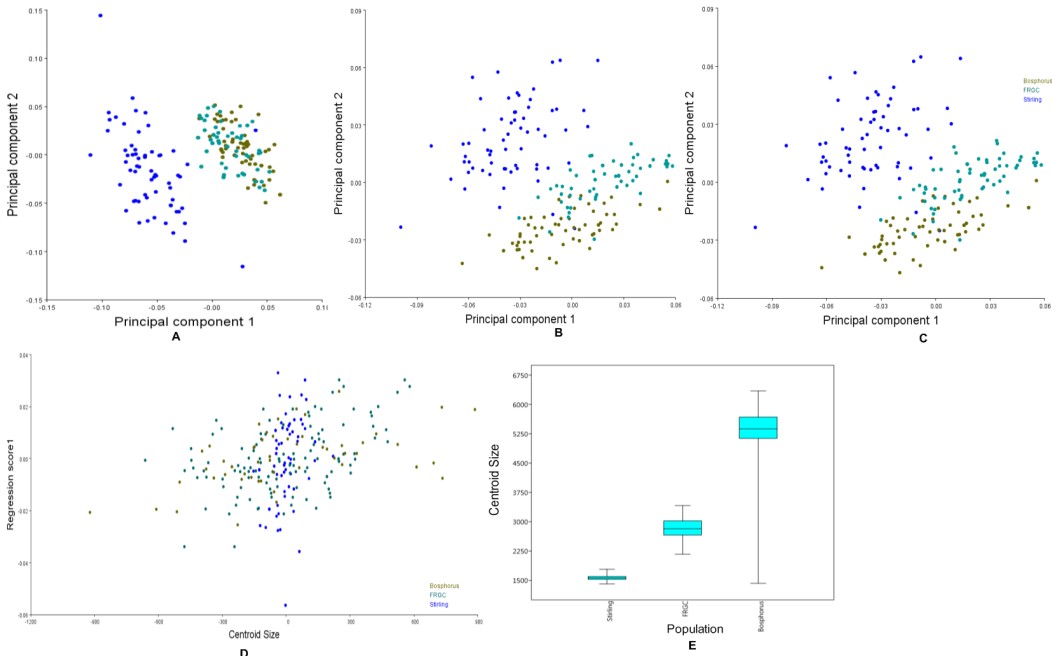

**Figure 6** **Principal components, multivariate regression and boxplots for the population groups.**
(A–C) Scatterplots of PC1 vs. PC2, which together explain more than 50% of the variance. (A) Manual anatomical landmarks. (B) Sliding semi-landmarks. (C) All landmarks. (D) Allometric regression of population mean shape. (E) Boxplot for centroid size for all populations, showing the size variation after averaging faces within populations.

**Table 3** **MANOVA results in terms of Wilks' Lambda.**

| Effect | Wilki | df1 | df2 | F | P |
|---|---|---|---|---|---|
| Population x CS | 0.0291 | 40 | 456 | 55.43 | <0.000 |
| Population | 0.0139 | 42 | 454 | 80.69 | <0.000 |

iteration, including data saving after sliding (*Botton-Divet et al., 2015*). This means that the workflow complexity is reduced, and the computational efficiency is higher in Viewbox than in Edgewarp. Although Morpho has been proven to be more efficient, since its functions can often be run on several cores many times, difficult coding is required to perform a single task.

Our analyses demonstrated significant differences in the average shape of the symmetric components across groups and the symmetric components of shape between individuals using the proposed method. The results of the Procrustes ANOVA suggested a modest but nevertheless appreciable variation in shape and size. Shape differences were statistically significant even after averaging faces within populations, and the small measurement errors (MAL = 0.063, SSL = 0.026, and CL = 0.027) show that the landmarks can be annotated with precision using the proposed method. Sliding semi-landmarks produced a more accurate error result, while manually placed landmarks gave a less accurate error value. This could be a result of difficulty in pinpointing the locus of the landmark (*Webster &*

*Sheets, 2010*). However, many approaches are available for addressing the measurement error when combining samples from multiple sources. A full discussion of this topic is beyond the scope of this study, but extended details can be found in (*Fruciano, 2016*). Our approach simply allows us to say that when the template used is changed, the shape does not change significantly (compared to other biological and non-biological sources of variation). To investigate allometry, a scatterplot of the regression score was plotted to show the regression of shape onto size pooling within populations (the projection of shapes in the direction of the vector of regression coefficients) vs. centroid size (*Drake & Klingenberg, 2007*). Box plots of centroid size differ significantly within each group. The centroid size was not log-transformed, as this transformation made no appreciable difference in the results. Despite their significance, the sizes of the effects being tested are small or similar in relative terms, and we must, therefore, interpret these effects with caution (*Daboul et al., 2018*). The MANOVA test based on Wilks' Lambda showed a significant result for both slope and intercept ($p < 0.000$). The visualization of face shape was done using PCA after the Procrustes fit of the dataset object; however, we present only axis 1 vs. axis 3, as the other two axes do not represent a normal facial shape. The results of visualization showed that Bosphorus faces have less prognatic faces and dropped chins, the FRCG faces have wide noses and long foreheads, and the Stirling faces have wide noses and wide foreheads.

In general, the face shapes in the Stirling and FRGC dataset are smaller than those in the Bosphorus dataset, and were characterized by geographical locations. This matches our expectations, as we were comparing samples from two neighboring regions and a different region. Both the Stirling and FRGC populations were based on the Western hemisphere (the UK and US, respectively) while the Bosphorus population was based on the Eastern hemisphere (Western Asia). Although an investigation of the variation among different data sources could be much more complex and complicated than that presented here, our goal in this study was to make the analysis as simple as possible, and this could be extended in future work. Readers should therefore be aware that the differences found between datasets are at least in part due to differences in the way these datasets were acquired (although it is hard to say how far this applies).

## CONCLUSIONS

This method combines pragmatic solutions for configuring an optimized pipeline for a high-throughput homologous multi-point deformable 3D facial signature. We warped only the reference surfaces and curves to each sample face, using an automatic homologous warping approach. The results of a Procrustes ANOVA show that the measurement error may be a source of substantial variation when combining different morphometric datasets, and may sometimes have an unexpected effect on parameter estimates (*Fruciano et al., 2017*). High-throughput phenotypic facial data such as these may be valuable in forensic studies of human facial morphology, anthropology, disease diagnosis and prediction, statistical shape or image analysis, face recognition, age estimation, facial-based sex dimorphism, and facial expression recognition. A limitation of the Viewbox is that it only runs on Windows OS, while Edgewarp runs on both Linux and Windows OS.

## ACKNOWLEDGEMENTS

We would like to thank the Face Recognition Grand Challenge, NIST, Bosphorus (Bogazici University), and Stirling/ESRC (University of Stirling) for prompt agreement to use their datasets, and the Computer Laboratory of the Faculty of Computer Science & Information Technology, Universiti Putra Malaysia.

### Funding

This work was supported by Putra Geran UPM (No. 9538100) and Fundamental Research Grant Scheme (No. 5524959). The funders had no role in study design, data collection and analysis, decision to publish, or preparation of the manuscript.

### Grant Disclosures

The following grant information was disclosed by the authors:
Putra Geran UPM: 9538100.
Fundamental Research Grant Scheme: 5524959.

### Competing Interests

The authors declare there are no competing interests.

### Author Contributions

- Olalekan Agbolade conceived and designed the experiments, performed the experiments, analyzed the data, performed the computation work, prepared figures and/or tables, and approved the final draft.
- Azree Nazri conceived and designed the experiments, performed the experiments, analyzed the data, performed the computation work, prepared figures and/or tables, data acquisition and preprocessing, and approved the final draft.
- Razali Yaakob analyzed the data, prepared figures and/or tables, and approved the final draft.
- Abdul Azim Abd Ghani analyzed the data, prepared figures and/or tables, authored or reviewed drafts of the paper, and approved the final draft.
- Yoke Kqueen Cheah analyzed the data, authored or reviewed drafts of the paper, and approved the final draft.

### Data Availability

The step-by-step methodology was fully implemented in Viewbox software which has now been separately cited in the article and all the raw data (3D digitized points and PCA scores) are available as Supplementary Files.

The landmark data is available in the Supplemental Files. All analysis was performed in PAST and morphoj. Raw image data are face datasets which require permission to be made available which can be requested here from FRGC, https://cvrl.nd.edu/projects/data/#face-

recognition-grand-challenge-frgc-v20-data-collection; Stirling, http://pics.stir.ac.uk/ESRC/; and Bosphorus http://bosphorus.ee.boun.edu.tr/HowtoObtain.aspx.

## Supplemental Information

Supplemental information for this article can be found online at http://dx.doi.org/10.7717/peerj-cs.249#supplemental-information.

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
