# Peer review of "Landmark-based homologous multi-point warping approach to 3D facial recognition using multiple datasets"

_PeerJ Computer Science, doi:10.7717/peerj-cs.249_

## Round 0.1 · original submission · Major Revisions

Dear authors, the reviewers are all in agreement about the necessity to strongly improve this work. Please, try to address all the reviewers requests.

Reviewer 1 ·

Basic reporting

There are problems with the language and with the level at which details are reported. Please, see comments to the Authors.

Experimental design

There are many ambiguities on what exactly has been done. Please, see comments to the Authors

Validity of the findings

Presently, the validity of the findings is, at best, hard to evaluate.

Additional comments

The manuscript “Automatic homologous multi-points warping for 3D facial landmark” by Agbolade and colleagues reports on a new method to automatically place points on 3D surfaces – and more specifically on digital surfaces of human faces.

The manuscript is potentially interesting for this journal and for the community, but its current state should strongly discourage its acceptance. There are three main problematic areas: 1. the English language is often poor, 2. what has been done is not described well enough to gauge the novelty of the method and whether the results are valid, 3. existing methods, particularly geometric morphometric methods, are at best mischaracterized (if not entirely misunderstood). I will comment on the specific aspects below.

The Authors insist in the Abstract (lines 21-23) and elsewhere (e.g., lines 48-51, 100) that manual annotation is more error-prone than automated methods. In the Abstract they also mention a series of other limitations. However, I do not think they provide substantial evidence for their claims. As far as I am concerned, I consider manual annotation like the current “gold standard” relative to which automated methods are usually tested. Typically, automated methods are considered in research workflows simply because they are less time-consuming than manual methods, not because they are better.

L58-69: The definitions of landmark types provided by the Authors contain many imprecisions and mischaracterizations and do not correctly reflect the original subdivision by Bookstein. For instance, the definition at lines 59-61 of type I landmarks is entirely not understandable. I suggest the Authors to re-read the relevant classical papers by Bookstein. In addition, they may want to consider reading the book “Geometric Morphometrics for Biologists” by Zelditch and collagues and perhaps also a recent discussion of various landmark classification by Wärmländer, et al.
The Authors’ misunderstanding/mischaracterization of current state-of-the-art methods with regards to landmark/semilandmark definition and placement resurfaces around lines 212-220, when invoking “100% homology” for semilandmarks, claiming that after sliding semilandmarks become homologous points (this is also claimed at lines 253-254), and that semi-landmarks are constructed by randomly spreading points across a surface based on landmark points and that they contain “progressively less biological information”. In addition to consulting again the relevant literature, the Authors should also consider that in current practice and parlance in geometric morphometrics semilandmarks are never homologous (they do retain positional correspondence, though) neither before nor after sliding (as sliding cannot make them homologous). Simply, after sliding they are treated the same as the homologous landmarks with respect to downstream analyses. Further, semi-landmarks are not typically constructed “randomly spreading points”. Rather, they can be, for instance, manually placed, or placed based on some criterion (e.g., equally distant from each other). However, there is no strict one-on-one correspondence between how these points are taken and the fact of being semi-landmarks.

The Authors should also be clear about what is novel in their method and what is the same as previous methods. This is far from obvious to me as various formulas and steps refer to existing practice and methods developed, for instance, by Bookstein and collaborators. I guess that perhaps the novelty is that semilandmarks are used to deform the actual surface (step 4 of the algorithm at lines 311-327), but, even if this is the case, it should be made explicit to the reader.

L289-291: the Authors claim that there is “no consensus” about how to quantify geometric morphometric data and cite a study which presents the difficulty of identifying landmark-specific error (i.e., some specific problem) to paint a grim picture of the current state of the field. They should perhaps consult the relatively recent review by Fruciano (2016), which discusses some of these methods and how there is some substantial consensus on them. Briefly, the methods which are practically speaking more useful are the ones which quantify globally the error (rather than at individual landmarks) and relate it to biological variation (rather than quantifying absolute error).

L294-300: the whole definition/explanation of Procrustes distance is problematic, for instance with respect to the reference the Authors make to removing “the effect of size” (what about rotation and scale?) and to “the first PC” (what about the others?). I encourage the Authors to consult carefully the literature on the subject. A somewhat similar problem surfaces again at line 327 where “average landmark error is computed using Procrustes distance”. But, if Procrustes distance is a distance between two shapes, how exactly is it used to compute “average landmark error”?

It is not entirely clear what the Authors have done in their study. For instance, the empirical datasets are not clearly described for the general public and it is not clear how one quantifies error with them. Is there a “true” shape that can be used to see how the new algorithm performs? From the way the manuscript is phrased it (for instance in the Results) seems like the Authors have repeated multiple times the landmarking with their method and found that running their algorithm multiple times on the same data gave similar results. If this is the case, obviously this is far from being a proper quantification of error. First, one would expect that two runs of the same algorithm on the same data should give the same results (or very similar ones). Second, both runs might be wrong with respect to the true shape that one wants to estimate. The Authors claim (lines 379-381) that manual landmarking is impractical when dealing with many points. This is true. But, then, with respect to what did they compute their error? Further, it seems that the sliding has been only used to place the semilandmarks on the surface but then it is not clear how shapes were compared. In current practice, one first places landmarks and semilandmarks on a surface (whether using automatic or manual methods, or a mixture of the two), then landmarks and semilandmarks are subjected to a generalized Procrustes analysis with sliding of the semilandmarks, finally the configurations of points thus aligned are subjected to downstream analyses. If the Authors have departed from this typical workflow, extensive explanations and clarifications should be provided.

Similarly, it is not clear how the “significance” of individual landmarks (discussed starting at line 339) was obtained nor what does it mean for landmarks to be “significant”. I should perhaps also note that after generalized Procrustes analysis and/or sliding of semilandmarks the points are not independent from each other, so it is often impossible/impractical/inappropriate to compute landmark-specific error.

L372-374: I am not clear why the Authors report the definition of centroid size here.

L396-397: the Authors seem to refer to different datasets used here as different species (which seems to be echoed in the table presenting the Procrustes ANOVA results). However, I had understood that all these datasets are data from human faces, are not they?

L413-416: the point that combining different datasets/devices could introduce error has been made in earlier studies (e.g., Fruciano 2016; Fruciano, et al. 2017) , which should be cited here.

L417-421: t tests and “feature” appear in this part, but it is not clear what they refer to.

Table 6: the term “Repeatability” has a precise use in current geometric morphometrics (see Fruciano 2016) and should be avoided here (unless the Authors actually compute repeatability, that is).

English problems
The manuscript suffers from many problems with the English language. I will present a few examples below but, please, do not consider this an exhaustive list as there are many more. The Authors should go through the manuscript and make sure they fix the English before resubmission, enlisting the help of a native speaker if necessary (as it seems the case).
L42: “is a three-dimensional object”
L58: “into three types: “
L70-71: the whole “anchor points on a face graph” thing is unclear/unknown to me
L101: “landmarking”
L115: “landmarks that fulfil”
L120: I suggest “further” instead of “furthermore”
L150: I suggest to remove “homologous” (as I doubt anyone can find 500 homologous points on a face, if we use biological/operational definitions of homology normally used in geometric morphometrics).
L167,172 “the Stirling”, “the Bosphorus”
L185 “description”
L198: maybe the Authors mean “overlapping” instead of “overlapped”?
L330-337: the whole section is not understandable

Minor problems:
L30-32: these results can be omitted from here as these datasets and their analyses are not presented in the abstract.
L71-73: rather than stating the obvious that no human face is ever two-dimensional, the Authors could perhaps refer to Cardini’s work (Cardini 2014) about the potential problems arising from a 2D projection of a 3D shape.
L92: do the Authors actually mean that this other method requires little investment of time (which would be a good thing) or do they mean that it requires substantial time (as the phrasing seems to imply)?
L200: the use of the term “morphological integration” should be avoided because in biology it has a very different meaning (i.e., non-independence between parts)
Not all symbols of formulas are explained (for instance the formulas at lines 232-234).
L329 a space is missing

References cited in the review
Cardini A 2014. Missing the third dimension in geometric morphometrics: how to assess if 2D images really are a good proxy for 3D structures? Hystrix, the Italian Journal of Mammalogy 25: 73-81.
Fruciano C 2016. Measurement error in geometric morphometrics. Development Genes and Evolution 226: 139-158. doi: 10.1007/s00427-016-0537-4
Fruciano C, et al. 2017. Sharing is caring? Measurement error and the issues arising from combining 3D morphometric datasets. Ecology and Evolution 7: 7034-7046. doi: doi:10.1002/ece3.3256
Wärmländer SKTS, Garvin H, Guyomarc'h P, Petaros A, Sholts SB Landmark Typology in Applied Morphometrics Studies: What's the Point? The Anatomical Record 0. doi: doi:10.1002/ar.24005

Reviewer 2 ·

Basic reporting

The quality of language should be improved. Sometimes, the text is hard to follow and
it contains a lot of typos or grammatical errors which are making the understanding
even worse.
The intro section would benefit of further structuring. For example, the related work
is tossed in the introduction section and starts without further warning. The methods
mentioned in the related work "section" lack logical structuring. One method even
describes a video-based approach, unlike the others, so its hard to understand why it
should be relevant, etc.
Figures numbering is shifted starting after the Figure 3 (probably because Figure 3 is
used twice and then the counting continues).
Raw data are provided, however, the comprehensive description is missing.
The authors are criticizing manual annotation and later, they admit they are also using
it for annotation of the most important landmarks.

Experimental design

There is a typo in Eq. (1) the second order partial derivatives of h should be over x, y
and z, not only over x. The Thin-plate Spline acronym TPS is used without first being defined. There are missing spaces when inline mathematics is used, which makes it harder
to read. I am quite puzzled of measurements in "mm", firstly because this would make the
measurements incomparable, e.g., adults and children and secondly because it would require some aditional callibration which is not mentioned anywhere in the text.
The section describing the landmark error measurement contains a lot of stuff which fits into the related work, it is hard for the reader to realize what is the metric used in the author's experiments.
From the description, as it is now, it is not possible to replicate the author's results.

Validity of the findings

The novelty of the method is not specified. The reported results look suspiciously good in
comparison to the state of the art. Some discussion why is that would be more than welcome. The tables 2, 3, and 4 could be presented within one table for easier comparison.
It is not clear, why the authors propose to introduce so much semi-landmarks when only 16 of them carry some important information, moreover, only these 16 landmarks are annotated manually.

Additional comments

The paper topic looks interesting. However, there are several issues, which prevents me from accepting the paper in its current form. The quality of English and paper structuring should be improved. Some parts are hard to follow because of grammatical errors and typos, some would benefit from better structuring.
The reported results look suspiciously good in comparison to the state of the art methods. The discussion of why this happens would be welcome.

Reviewer 3 ·

Basic reporting

This paper proposed an Automatic Homologous Multi-Points Warping (AHMW) for 3D facial landmarking.

The writing is not readable.
Even for abstract, there are some typos.

'wether' face recognition --> whether
'insuffient' landmarks --> insufficient

Please polish up the writing for the whole paper.

Experimental design

There are some experiments on three datasets using 500 landmarks.

Validity of the findings

The experimental results show that the method is robust when compared with the state-of-the-art methods with minimum localisation error (Stirling/ ESRC:0.077; Bosphorus:0.088; and FRGC v2: 0.083).

Additional comments

Please polish up writing to get the paper accepted.

---

## Round 0.2 · Major Revisions

The are still some problems related to methodology and in making claims which are either hard to evaluate or unsupported by evidence. Please try to address the considerations of reviewer 1.

Reviewer 1 ·

Basic reporting

Serious problems with the English and it is not entirely clear what the Authors have done. Please, see section "General comments for the author" for examples and in-depth discussion.

Experimental design

There are (still) many methodological problems, including that what the Authors have done remains elusive. Please, see section "General comments for the author" for examples and in-depth discussion.

Validity of the findings

Because it is not clear what has been done exactly, findings and claims are hard to evaluate. For my current interpretation of the results, as they are provided, the claim of small error with this new method does not seem to be supported. Please, see section "General comments for the author" for examples and in-depth discussion.

Additional comments

I have reviewed the revised version of the manuscript “Automatic Homologous Multi-Points Warping for 3D Facial Landmark” by Agbolade and colleagues.

I have to commend the Authors for making an effort to accommodate my comments in the revised version of the manuscript, and for politely addressing my concerns. The text has also overall been improved.
Having said that, I think that the manuscript is still far from being publishable and a series of problems remain. Line numbers below refer to the PDF version of the manuscript

L23: Again, I appreciate that we might not have the same opinions when it comes to manual landmarking, but the claims that “current facial landmark methods in 3D […] contain insufficient landmarks” and “lack homology” are not supported by any evidence. Current approaches, when digitizing only a few landmarks such as in many geometric morphometric studies, often insist on homology. Other approaches relax the assumption of homology (for instance by using semilandmarks) and allow to place very many landmarks. This statement should be qualified more precisely and made more fitting to the manuscript and the novelty of the method.

The whole section at lines 71-88 is hard to understand and the concepts are not clear. For instance, at line 83 a reference to a “shape parameter” is made, as if this is some sort of scalar or something that is totally independent from the thin-plate spline and that is “estimated” through Procrustes superimposition (line 84). However, the Procrustes superimposition is simply a transformation which removes location, orientation and scale and allows one to analyze shape (which are the coordinates after the superimposition). Please, rephrase this section.

While I do appreciate that the Authors have consulted the relevant literature on landmark types, the explanation (L100-120) is still extremely difficult to follow. It should be rephrased. This problem is somewhat in-between a problem with English (see below) and a problem with conceptual representation of these landmark types. The Authors are very welcome to find their favourite way of rephrasing the text, but I would suggest that perhaps simplifying and clarifying the definitions (one sentence per type) and forwarding the reader to the relevant literature might be the easiest choice to implement.

L227: The sentence “the method was fully implemented in ViewBox 4.0” is not clear. Have the Authors implemented their method for automatic point placement in ViewBox or have they used ViewBox for the geometric morphometric analyses described? Please, clarify.

I thank the Authors for clarifying their methods and what the novelty relative to work by Bookstein and colleagues is (L26-29; 168-172). Nonetheless, it is still not clear what is novel in this manuscript. For instance, methods to project a set of points from a template to all surfaces in a sample, as well as for sliding of semilandmarks, relaxation on surfaces and mesh deformation already exist, are at least in part implemented in the literature, and are regularly used. For instance, see functions placePatch and relaxLM (and other related functions) in the R package Morpho. I encourage the Authors to stress even more what the actual elements of novelty of this are, and clearly distinguish these from existing/implemented methods.

L304-312: The explanation of principal component analysis is probably unnecessary – or at least it is unnecessary in biology (I am not sure about computer science). If the Authors decide to keep it in, they should make it correct and include more details. For instance, was the eigenvalue decomposition performed on the matrix M or, as I assume, the matrix K? The description also sounds like “eigenvalue decomposition […] produce highest ranking eigenvectors […] with the help of […] eigenvalues”. However, the eigenvalue decomposition produces a set of eigenvectors and eigenvalues, it is not that one is obtained with the help of the other. Perhaps the Authors refer to the ranking of principal components here. Further, it is not clear how many principal components were retained (here x) and using which criterion.

L314-336: The title should be “Measurement error”. Also, there are a series of imprecisions and lack of clarity which should be improved. For instance, it is not clear why it is more difficult in geometric morphometrics compared to traditional multivariate morphometrics. I do agree that non-independence of landmarks create issues for certain analyses (i.e., landmark-specific computations of error, as presented in the previous version of the manuscript), but, if data is treated as multidimensional, I do not see very large differences between traditional multivariate morphometrics and geometric morphometrics in terms of analysis of measurement error. Further, Procrustes ANOVA is not based on generalized Procrustes analysis. It uses data which has been subjected to generalized Procrustes analysis/Procrustes fitting and exploits the fact that these are mean-centered, but Procrustes ANOVA itself is not based on generalized Procrustes analysis. Please, see Fruciano 2016 or the original sources for details on this and rephrase these parts accordingly. Also, it is not clear why dimensionality reduction (L334-336) was performed here, as Procrustes ANOVA can be performed using the Procrustes-aligned coordinates.

As before, it is not clear how the measurement error analysis was performed. From the way this is phrased at lines 340-356, it seems like a single mesh was used as template and here 16 fixed points were digitized. Then the method described above was used to automatically place and slide points/semilandmarks. Then the Procrustes distances were computed between target and reference for Procrustes ANOVA. However, it is not clear when/how manual annotation of five specimens by the same person (L320) comes into place. Also, no Procrustes superimposition is ever mentioned (however, if this has been carried out in MorphoJ, it has most likely been performed as it is an essential step after importing data). For the reasons above, it is not clear what the repetitions represent here. And what the “error” represents or whether this “error” is actually variation due to rotation, translation and scale (which would be present if Procrustes fitting were not performed). To be clear, if the Authors performed the analysis to test how the choice of a manually-annotated template affects variation in shape in target shapes they should do the following:
1. Chose 5 (or whatever other number) surfaces as references
2. Use a sample (for instance 20 or 30, or entire datasets) of other surfaces (distinct from the 5 references, each one belonging to a different individual)
3. Perform their method of automatic landmarking using the same sample from point 2 but each time using a different reference from point 1
4. Superimpose all the configurations of points obtained at point 3 (and not the references) in the same Procrustes analysis
5. Perform Procrustes ANOVA on the superimposed configurations from point 4 using Individual as factor (and possibly other factors, if they wish/are available)
The analysis described above would give information not about “error” in general but about the sensitivity of the method developed here to reference choice. This should be stressed very clearly all along the manuscript, as different sources of error exist and any claim about "error" should qualify what kind of error refers to. Importantly, as mentioned in my earlier review, the fact that this method produces consistent estimates of shape is a good thing but they may be “consistently wrong”. So, if there is no “ground truth” (e.g., manual annotation of the same surfaces) to which this can be compared, this should be made extremely clear to the reader.
To be clear, what I detailed above is just an example of what could be done in terms of studying measurement error, but I am not sure what the Authors have actually done because the way the manuscript is written does not allow me to figure it out.

L359-361: Please, document in more detail what was done with which software (also, this should be part of the Materials and Methods, not the results).

While the manuscript and the analyses presented are certainly much improved relative to the first version, I am not quite convinced by some of the figures and related analyses. For instance, I am not sure why the asymmetric component is presented in Fig. 5 A-D. This is not clearly described in the text and I am not sure which function does it have. Most importantly, if the PCAs have been performed only on the symmetric component – which would be reasonable – this should be mentioned in the text. Interestingly, typical analyses which include asymmetry in MorphoJ will then include a Side and an Individual x Side term in the Procrustes ANOVA, but these do not appear in Table 3. This is not to say the analysis is not correct, as it is really hard to evaluate at this stage. But the Authors should more clearly specify how exactly they have carried out the analysis. The issue with asymmetry appears again at lines 422-426, where differences between populations are discussed in terms of asymmetries and not in terms of their mean symmetric shape (although Table 4 does not mention asymmetry).
Further, I am not quite clear why centroid size in Fig. 6 B takes negative values and it is not clear how the Authors have used their data (supposedly replicates) for the comparisons of group means (tests based on Procrustes and Mahalanobis distances).


More importantly, the results in Table 3, as they stand now, show a very large error. It is not clear what exactly the Authors have done (see above the issue with Procrustes ANOVA and what the Residual term stand for), so I will have to suspend the judgement for now. But if the “Residual” term were variation between replicates of the same individual (as it is normally the case in this kind of analysis of measurement error), this accounts (based on the numbers provided) to 46% of total variation and accounts for more variance than any other supposedly biological source of variation (Population and Individual). The mean squares for the Residual term are also very similar to the mean squares for the Individual term. In other words, this means that (supposed) replicates of the same individuals vary more than the variation between individuals. If the final aim of this method were facial recognition in humans (supposedly within populations), this would mean that the error in landmark placement is so large to make the method unsuitable for this goal (but perhaps still useful when the biological signal is large, for instance when comparing different species). Again, I will suspend the judgement for now as it is not clear what exactly it is shown here.

Because of the lack of clarity in many parts and the possibility I just mentioned that measurement error is actually quite high, at this stage I will not evaluate the claims in the Discussion (e.g., the claim of small error, at lines 439-440).

The manuscript has still enormous problems with the quality of the English language. Even in the abstract, there are mistakes or the text is not clear. I will make here just a few examples of the problems. The Authors, however, are strongly encouraged to consult with a native speaker. To be clear, what follows are just examples, and only on the first page or so of manuscript, and much more should be done in the text. I do realize this might be annoying, but unfortunately, even if/when the other problems in the manuscript were to be fixed, I would not be able to suggest acceptance if the English is in this state.
Title: perhaps “landmarking” instead “landmark” would be more appropriate (same at L24).
L19-21: Apart from being unclear what the Authors mean, it is not appropriate to say that “scientists […] have been digging deep into research to know whether face recognition is performed holistically […]”. I mean, scientists are the ones using face recognition and developing methods for it. So it is not that they have to dig so deep in research. Perhaps the Authors refer to the fact that scientists have been asking whether one approach is better than the other?
L27: I am not clear what “concept” refers here to. Maybe “approach”?
L35: “Landmark-based” (without “The”)
L38: what does “listed” refer to? Also what about “collaborative behavior”? It is not very clear what the Authors mean here.
L42: “facing 2D face” does not sound quite right, maybe “affecting”?
L45: maybe “a drop”?
L52-53: have these methods received sporadic (i.e., rare) interest or is this one of the most vigorous research areas in biometrics? Cannot be both…

Minor issues:
L66-67: “geometric morphometrics” is not necessarily “the latest approach”, particularly if one includes in this field methods not based on landmarks but on function-fitting (e.g., elliptic Fourier analysis), which are used in biology since the early 80s (whereas landmark-based approaches, while older, had a strong impulse in the early 90s, so it is hard to frame them as particularly new). Most importantly, the description which follows refers only to landmark-based geometric morphometrics. I suggest avoiding reference to “the latest approach” (it is not that recent) and mentioning how popular they are, further mentioning directly “landmark-based geometric morphometrics”.

L364-366: It is perhaps worth to notice that the number of PCs found is due to the sample size.

L391-403: the description of methods belongs to the methods section (where certain analyses like the Kruskal-Wallis test are not even mentioned), not the Results.

Reviewer 2 ·

Basic reporting

The authors significantly improved the quality of the writing and addressed all reviewers comments in the updated manuscript.

Experimental design

The experimental evaluation was improved by using different metric, supported by a clearer description.

Validity of the findings

All underlying data have been provided.

Additional comments

The overall quality of the manuscript is significantly improved, thanks to addressing all reviewers comments. Now, I do not see any issue preventing the publication of the manuscript.

Reviewer 3 ·

Basic reporting

The revised version has solved the problem I mentioned before.

Experimental design

The experiment is extensive.

Validity of the findings

The proposed method is novel and practical.

Additional comments

Please have a further improvement on the writing.

---

## Round 0.3 · Minor Revisions

I have evaluated the paper and I feel that after these revision rounds the 'scientific' issues are sufficiently addressed and the remaining comments from Reviewer 1 are mostly related to language and comprehension. If the communication issues were to be addressed then the paper would be of much greater value to a broader audience. Therefore, I would like to give you this final revision opportunity to edit the language before I recommend Acceptance.

Reviewer 1 ·

Basic reporting

see below

Experimental design

see below

Validity of the findings

see below

Additional comments

I have reviewed the new version of the manuscript “Landmark-based homologous multi-point warping approach to 3D facial investigation of multi-datasets” where the Authors describe a new/improved method of placing (semi)landmarks on 3D surfaces of human faces.

I have to say I have mixed impressions about this new version. On one hand, I have to commend the Authors for the effort they put in and I have to say that the manuscript has certainly been improved. On the other hand, however, it seems this manuscript is not improving at the rate I was hoping for, and it seems still far from being acceptable for publication.

Differently from previous reviews where I have been very detailed, here I will provide just a few pointers, and I invite the Authors to go back to my last review, as various issues are still there and need further clarification/rewriting/improvement.

What is new here compared to what is existing is still not 100% clear to the reader. This should also be incorporated in the Abstract.

A series of statements are still unwarranted and unsupported.
For instance, at lines 51-52, a claim is made that “the complex and time-consuming computational process has made research into 3D facial landmarks generally unreproducible.”. This is a very strong statement, not very well supported by evidence. The Authors, if they want to make such claims, should refer to actual results, where reproducibility (or similar parameter, such as repeatability) is quantified, citing the relevant research.
Similarly, at lines 285-288 a suggestion is made that by using projection and iterative sliding “Using this method, the issues of flawed data are automatically handled” and that “This eliminated the hassle of preliminary imputation of missing landmarks”. This, at least in the way it is written, is not correct. Normally, landmarks are missing because a certain structure is missing from the data, and no sliding is going to take care of that per se.
Again, these are just examples.

Some of the analyses are not adequately/properly described.
For instance, at lines 353-354 a statement is made that “The statistical significance of pairwise differences in mean shapes was assessed in 1,000 permutations using the Procrustes distance by a pooled within-group” and this does not correspond to any analysis I know of. Probably, the Authors are conflating in the same sentence a test of differences based on permutations and Procrustes distances with a CVA, which uses pooled within group covariance and is described in the next sentence. Similar unclear references to “pooled within group” are made elsewhere (L418-419).

As I mentioned before, here what the Procrustes ANOVA/analysis of error is saying is not clear, as the Authors do not describe exactly how the analysis has been performed.
Earlier, I had given a suggestion of one way of carrying such an analysis. Now, from the section at lines 316-319 is not clear whether what I had suggested is exactly what has been done. If it is, the Authors should clarify that the same individuals (apparently 5 per dataset) were acquired nine times each using the three templates.
Importantly, if this is what they have done, the Authors should be clear when discussing what their results mean. That is, this technique simply allows to say that changing the template used, the shape does not change that much (compared to other biological and non-biological sources of variation). This does not mean that the method is generally speaking robust (as implied in the Abstract).
Similarly, the Authors should warn the reader that the differences found between datasets are at least in part (hard to say how much) due to differences in the way these datasets were acquired.

I should notice that for all purposes in biology a significant allometric effect accounting for 13.28% of total variation (lines 435-436) is far from negligible.


Further, I am not clear about whether all material has been updated.
For instance, Figure 1 still reports “average landmark error” and “bias” which have been removed from the text, and Figure 5 still refers to asymmetry.
I also note that the “track changes” version the Authors have provided for review was not showing all the changes from the previous draft (in fact, it was showing very few compared to the ones made).



There are still enormous problems with the English language and, to reiterate, even if the rest was perfect, I would not be able to suggest acceptance with such an English language.
I do understand that not being native speakers creates a challenge. However, too much is lost in translation for a reader to actually understand what the Authors mean.

In their reply to my comments, the Authors wrote “All necessary grammatical blunders have been corrected before we finally handed over the manuscript to professional native speaker editor.”. Well, whoever was this “professional native speaker editor”, he/she did a very poor job, as even typos and misuse of English language are still present. For instance, at line 161 “in three-dimensional” is used in lieu of “in three dimensions”. Or at line 122, where the Authors write “The approach demonstrated suitability cases”, probably meaning something like “The approach proved suitable in cases”.

Most importantly, here the main issue is not typos or grammar, but the fact that what the Authors actually mean is not understandable. This is why I suggested a native English speaking colleague: this must be someone who understands what the Authors have done and mean. A few examples of this:

- The sentence at lines 25-27, in the Abstract, is not clear as a reference to “datasets” is made; do the Authors mean “each of the target objects in a given dataset” when describing their method? (as this is the part where how the method works is presented); similarly, what does “and the workflow is compared” means in the context at line 29?

- At line 40, the Authors write “Besides the pose-invariant and illumination challenges”, but in the previous sentence the change of pose is presented as a challenge. Considering that “being pose-invariant” means that “pose does not matter”, it is not clear what the Authors mean. Probably, they meant “Besides variation due to pose and illumination affecting 2D face data” (or something along these lines)

- The sentence at lines 156-159 is not clear; probably the Authors refer to using a simpler workflow, compared to the one they cite, by performing an existing step in Viewbox; but the way this is written many interpretations are possible

- The Authors claim that “Our analyses demonstrated significant differences in facial symmetry” (L463) but this is not what – I think – the Authors have done as they have compared the average shape (of the symmetric component) across groups and the symmetric component of shape between individuals; this is quite difference from finding differences in symmetry (which would mean that one group is more symmetric than another, for instance)

In addition to this, there is a broader issue of how the manuscript is constructed, in that it does not follow a clear logic and path. This is particularly evident in the Introduction, where a series of statements more or less true (with some issues with interpretation and English language) are made. However, the order of these statements is not very clear or functional in terms of logic.

Again, all of this requires not simply changing a few typos but extensive re-thinking and rewriting in a simple and intelligible fashion (and following a clear logic when moving from one section to another) of what the Authors want to say, so that it is clear to the reader.

---

## Round 0.4 · accepted · Accept

All the reviewers requests are addressed